# Research Progress of Polyvinyl Alcohol Water-Resistant Film Materials

**DOI:** 10.3390/membranes12030347

**Published:** 2022-03-20

**Authors:** Baodong Liu, Jianhua Zhang, Hongge Guo

**Affiliations:** 1Faculty of Light Industry, Qilu University of Technology (Shandong Academy of Sciences), Jinan 250353, China; lbd980918@163.com; 2Institute for Sustainable Industries & Liveable Cities, Victoria University, P.O. Box 14428, Melbourne, VIC 8001, Australia; jianhua.zhang@vu.edu.au

**Keywords:** polyvinyl alcohol film, water resistance, modification method, application

## Abstract

Polyvinyl alcohol (PVA) is one of the few biodegradable synthetic resins from petroleum-based sources that can alleviate white pollution in the environment. PVA film materials have excellent properties, such as high barrier, high transparency, high toughness, biocompatibility, and adjustable water solubility. However, due to the presence of hydrophilic hydroxyl groups in the side chain of PVA resin, when PVA film is placed in a humid or water environment, swelling or even dissolution will occur, which greatly limits its application. Therefore, it is necessary to modify PVA resin to improve water resistance without reducing other properties and can also impart various functionalities to it, thereby widening the application range. This paper reviews the water-resistant modification methods of polyvinyl alcohol and the application of water-resistant films and provides an outlook on the development trend of PVA water-resistant films.

## 1. Introduction

Plastic film material is one of the most widely used materials that brings convenience to the development of human production [1,2]. With the progress of science and technology, the development of plastic film has been rapid [3], and at the same time, “white pollution” also brings serious harm to the environment [4,5]. General plastic film materials such as polyethylene (PE), polypropylene (PP), polyvinyl chloride (PVC), and other petroleum-based materials, with good barrier properties, high mechanical properties, transparency, and other characteristics, have a wide range of use, but at the same time also generate a large amount of waste, and will generally not automatically degrade in a short period of time. To decompose plastic film into simple harmless molecules, such as CO_2_ and H_2_O, takes a long period of even hundreds of years [6], which brings great harm to the environment [7,8].

Polyvinyl alcohol (PVA) is the only vinyl polymer that can be used by bacteria as a carbon source and energy source. Under the action of bacteria and enzymes [9,10,11,12], PVA can be degraded by 75% in 46 days, so it belongs to a kind of biodegradable polymer material [13]. PVA is a water-soluble polymer with a carbon chain as the main chain and a large number of hydroxyl groups in the side chain [14,15]. It can be produced on a large scale by non-petroleum routes, with low price and good film-forming properties [16,17,18]. Figure 1 shows the structural formula of PVA resin. PVA film is flexible, transparent, non-toxic, non-hazardous [19], and biocompatible; it has good mechanical properties [20], chemical resistance, and gas barrier, and can be completely biodegradable [21,22,23], in line with the development of green environmental protection needs. Therefore, replacing non-degradable polyethylene and polypropylene materials with environmentally friendly PVA film materials is a major trend in the future [24].

PVA resin is obtained by the hydrolysis of polyvinyl acetate (PVAc) [25]. In the process of alcoholysis of PVAc, part of the alcoholysis generates polyvinyl alcohol [26], so the physical and chemical properties depend on the degree of polymerization and alcoholysis, especially the degree of alcoholysis [27]. Figure 2 is a Schematic diagram of PVA and PVAc coexistence. Through the control of the preparation process, it can be made of different degrees of alcohol dissolution of different degrees of polymerization of polyvinyl alcohol resin (commonly used grades are PVA-1788, PVA-1799, PVA-2488, etc.—the first two digits are the degree of polymerization, the last two are the degree of alcohol dissolution) [28]. Through the solution of the cast film method [29], wet extrusion blowing film method [30], dry extrusion film method, and other film-forming processes to get different properties of PVA film. PVA films are generally divided into two types: water-soluble films made from polyvinyl alcohol resins with an alcohol solubility of about 88%; and insoluble films made from polyvinyl alcohol resins with an alcohol solubility of 98% and above. PVA film has adjustable water solubility, whether water-soluble film or insoluble film, the resistance to water is very low, and, at a certain time and temperature, can be dissolved in water; therefore, the application range of pure PVA film is narrow and needs to be modified for water resistance.

This paper reviews the water-resistant modification methods of PVA and the application of water-resistant films and provides an outlook on the development trend of PVA water-resistant films.

## 2. Water-Resistant Modification Method of PVA

When the PVA film is in a humid or water environment, it will swell or even dissolve; that is, it has poor water resistance [31] due to the presence of hydrophilic hydroxyl groups in the side chain of the PVA resin. Comparing general-purpose PE with PP film materials, there is still a gap, so the modification to improve the mechanical properties and water resistance of PVA film and other properties are needed [32]. The modification methods are mainly divided into two categories: single method modification and synergistic modification.

### 2.1. Single Method Modification

Single method modification is one method to modify PVA to improve the water resistance of PVA film, mainly nano-fill modification, polymer co-blend modification, and chemical cross-linking modification.

#### 2.1.1. Nano-Fill Modification

Nanomaterials have a large specific surface area, high surface polarity, high rigidity, good thermal stability, etc. [33]. Nanomaterials can be uniformly dispersed in the PVA matrix, changing the force between PVA molecular chains, forming strong hydrogen bonds, and changing the crystalline structure [34,35,36,37]. Nanomaterial filling modification not only improves the mechanical strength and water resistance of PVA film, but also further expands the functionality of PVA film [38,39,40]. Figure 3 shows a diagram of the nano-filling modification. Commonly used nanoparticles are silver nanoparticles (Ag NPs), nano-TiO_2_, nanocellulose (NC), cellulose nanofibers (CNFs), etc.

Sarwar et al. [41] prepared nanocomposite films by adding different proportions of NC and Ag NPs to PVA. The scanning electron microscope (SEM) results show that NC and Ag NP particles are uniformly distributed in PVA. Fourier transform infrared (FTIR) results show that pure PVA, NC, and PVA/NC show peaks at 3286.4 cm^−1^, 3291.1 cm^−1^, and 3259.6 cm^−1^, respectively, indicating the presence of free hydroxyl groups due to strong inter- and intra-molecular bonds. The interaction between the hydroxyl groups on the surface of the NC and the hydroxyl groups in the PVA matrix resulted in a slight change in the O-H tensile strength. The shift of the PVA hydroxyl peaks was not significant after the addition of NC and Ag NPs to the PVA matrix. Regular PVA bonds were observed in the spectra of PVA/NC 4% and PVA/NC 4%/Ag 1% films, indicating that the addition of nanocellulose and Ag NPs did not have a significant effect on the molecular structure of PVA. No significant chemical bonds were formed between PVA, NC, and Ag NPs. After the water vapor transmission test, it was found that the water vapor transmission was reduced with the addition of nanoparticles. WVTR of PVA 10 wt% is 43.09 ± 2.15 (g/m^2^ h), WVTR of PVA 10 wt%, 16 wt% NC is 37.39 ± 2.77 (g/m^2^ h), WVTR of PVA 10 wt%, 16 wt% NC, and 0.5 g Ag NPs is 14.36 ± 2.71 (g/m^2^ h); these results indicate that the incorporation of nanoparticles can improve the water resistance of PVA films.

Tang et al. [42] prepared antibacterial PVA films doped with nano-TiO_2_ by the solution blending-casting method, and the effects of different nano-TiO_2_ additions (0%, 0.5%, 1%, 3%, 5%, and 7%) on the characterization of PVA films were investigated. The SEM results show that nano-TiO_2_ is well embedded and dispersed in the PVA matrix as a whole, but when the content of nano-TiO_2_ exceeds 1%, some small agglomerates are formed. Compared with the pure PVA films, the swelling rate and water absorption of the films containing 0.5~7% nano-TiO_2_ were reduced by 22.82~81.79% and 3.59~10.7%, respectively.

Qiu et al. [43] treated grape skins ultrasonically to obtain ultrasonic grape skins (UGS) suspensions, in which the main ingredient after testing was NC, while biodegradable multifunctional composite membranes were subsequently prepared by combining UGS and PVA. UGS and PVA formed a good interface, which was attributed to strong hydrogen bonding. The crystalline nucleation properties of the films were further characterized by differential scanning calorimetry (DSC). The addition of UGS promoted the formation of a more stable and abundant crystal structure in the composite films. The crystallinity index (CrI) increased significantly after the addition of UGS. The CrI of pure PVA was 28.86%, and that of UGS15-PVA was 66.27%. On the basis of polarized light microscopy measurements, well-distributed bright areas were visible in the composite films with a higher density than in the pure PVA films. The water content of the film was tested in a 93% humidity environment. Pure PVA film was 21%, UGS5-PVA:20%, UGS10-PVA:15%, UGS15-PVA:12%, which confirmed the enhancement of water resistance of PVA after UGS was added.

Popescu et al. [44] prepared bio-nanocomposite films of cellulose nanocrystal (CNC)-reinforced PVA by the solvent casting method, and the test results showed the existence of hydrogen bonding interactions between PVA and CNC with changes in conformational rearrangement and the addition of CNC affected the microcrystal size and crystallinity. The water absorption capacity of the PVA/CNC films decreased from 93% to 75% with increasing CNC content, indicating the involvement of hydroxyl groups in the new hydrogen bonding interactions, demonstrating the improved water resistance of the films.

Sánchez-Gutiérrez et al. [45] prepared biodegradable films by incorporating CNF obtained from olive trees into PVA. SEM indicates a uniform distribution of CNF without layer separation; FTIR shows a peak at 3250 cm^−1^, which is attributed to the typical O-H stretching vibrations of inter- and intramolecular hydrogen bonds. After water vapor transmission tests, the results showed that the PVA-(L) CNF film showed lower water vapor permeability (WVP) compared to the pure PVA film (*p* < 0.05). The sharp decrease in WVP was evident in the 5% TB PVA-CNF film, which showed the lowest WVP value compared to the pure PVA film (6.97 × 10^−7^ g/s·m·Pa) (2.82 × 10^−7^ g/s·m·Pa) (*p* < 0.05).

#### 2.1.2. Polymer Co-Blend Modification

Polymer co-blend modification is conducted by adding a large number of polymers that can generate hydrogen bonds with the hydroxyl groups in PVA so that the hydroxyl groups of PVA and water affinity are reduced for water resistance purposes. Figure 4 shows the polymer co-blend modification.

The polymer commonly used for blending with PVA is chitosan (CS), and the -NH_2_ and -OH on the CS molecular chain make it possible to rely on intermolecular hydrogen bonding for uniform dispersion in the PVA film [46]; after blending with CS, the PVA film also has antibacterial properties. Liu et al. [47] prepared PVA blend films containing chitosan (CS) by a simple solution cast and electrospray method. FTIR showed that the band at approximately 1077 cm^−1^ indicates the presence of a hydroxyl group with a polymeric association and a secondary amide. The band at 1450 cm^−1^, appearing for weight fractions of 20% CS or more, was assigned to C=N pyridine ring vibrations. This confirms the formation of complexes between PVA and CS. Water vapor permeability (WVP) through the films was determined at 23 °C and 60% RH. The WVPs of PVA/CS-2 and PVA/CS-2.5 films (16.41 ± 2.66 g cm^−1^ s^−1^ Pa^−1^, and 18.03 ± 2.82 g cm^−1^ s^−1^ Pa^−1^) were higher than that of pure PVA film (15.81 ± 2.56 g cm^−1^ s^−1^ Pa^−1^). This indicates that the interaction between PVA and CS can increase the water vapor barrier properties.

Starch can be obtained from the waste streams and by-products of multiple renewable food sources, including corn and potatoes [48,49], and can be completely degraded in any environment [50]. After PVA and starch were mixed, the water absorption decreased compared to pure PVA due to hydrogen bonding between PVA and starch, which affected the number of free -OH groups [51,52]. Pantelic et al. [53] mixed thermoplastic starch (TPS) and PVA 1:1 to produce PVA/TPS film by extrusion. From the FTIR spectrum of the PVA/TPS film, the presence of a broad peak in the area of 3200–3500 cm^−1^ could be attributed to the vibrational stretching of hydroxyl (–OH) groups due to the inter- and intra-molecular hydrogen bonding of –OH groups in the PVA and starch. The results of the water absorption test showed that, in the starting 48 h of immersion, approximately 40% of moisture was absorbed, while a significant increase in moisture uptake was detected after 72 h of exposure (80%). After 72 h, the water uptake started to decrease, probably due to the dissolution of highly hygroscopic PVA/TPS material. Water uptake of less than 60% was calculated after seven days of exposure.

#### 2.1.3. Chemical Cross-Linking Modification

Chemical cross-linking modification is the formation of chemical bonds between PVA molecules to form a cross-linked network, thus improving the mechanical and water resistance properties of PVA films [54]. To improve water resistance, the hydroxyl groups on the PVA chains can be used to chemically react and transform into water-resistant groups, which can also improve the tensile strength and thermal stability of PVA films [55,56].

Using the hydroxyl group on the PVA chain and the organic acid or anhydride in high-temperature esterification reactions causes the introduction of carbonyl groups between the molecular chains of PVA [57]. Figure 5 shows the cross-linking reaction between boric acid and PVA [58].

Gao et al. [59] subjected stearic acid to esterification with PVA to esterify some of the hydroxyl groups in the PVA molecule. The results of the infrared spectroscopy test showed that PVA was esterified with stearic acid, and -OOC(CH_2_)_16_CH_3_ was introduced into the side chain of PVA; the results of the water resistance test showed that the water resistance of PVA increased with time. Huang et al. [60] prepared esterified PVA films by the flow casting method using PVA and maleic anhydride as raw materials. The tensile properties and water resistance of PVA films were also tested. The results showed that the tensile strength of the esterified PVA film was greatly increased, and the tensile strength increased to 49.6 MPa at 20% maleic anhydride content. Moreover, the water resistance was greatly improved, and the dissolution rate decreased by nearly 60% at 204% maleic anhydride content and 4 h dissolution. Suganthi et al. [61] investigated the effect of three different organic acids, malic acid (MA), tartaric acid (TA), and lactic acid (LA), on the physicochemical and biological properties of PVA. After contact angle measurements, the films became moderately hydrophobic by cross-linking of carboxylic acids compared to the highly hydrophilic nature of PVA, and the results showed that the contact angle of PVA/LA was the largest.

PVA acetylation modification is achieved by dehydration of the hydroxyl group of the PVA molecule with the carbonyl group of the aldehyde compound to form an acetal compound [62]. Figure 6 shows the cross-linking reaction between glyoxal and PVA.

Zhang et al. [63] cross-linked PVA with glutaraldehyde and urea by acetal, and finally determined the structure of the substance and characterized its properties by infrared spectroscopy FTIR, thermogravimetric TG analysis, physical and mechanical properties, and contact angle. The results showed that the water resistance and thermal stability of PVA films could be improved by cross-linking reactions of glutaraldehyde, urea, and PVA hydroxyl acetal.

### 2.2. Synergistic Modification

With the development of society, products for film performance requirements become higher, and a single modification of PVA to prepare the PVA film will not meet the application needs. Researchers have begun to focus on synergistic modifications. For example, taking polymer-blend modification as the main polymer/PVA matrix and then adding nanoparticles or cross-linking agents so that the PVA composite film not only retains the original superior performance, but also has new properties, such as water resistance, antibacterial properties, etc.; this creates a product superior in performance to that of a single modification to the PVA film.

#### 2.2.1. Nanofill and Polymer Co-Blend Modification

The nanoparticles are dispersed into the PVA blending solution as the reinforcing body, and hydrogen bonding occurs with PVA and the blends at the same time, which further reduces the affinity between free -OH and water in PVA on the basis of hydrogen bonding between PVA and the blends. Figure 7 shows a diagram of the nanofiller and polymer composite acting together.

Wu et al. [64] prepared coconut shell polyol (CP)/PVA biocomposite films and CCNF/CP/PVA films by blending CP with PVA and adding coconut shell cellulose nanofiber (CCNF) reinforcement, respectively. After comprehensive characterization, the CP/PVA films exhibited higher stability in the medium temperature range, while the CCNF/CP/PVA films had higher tensile strength, elongation at break, and water resistance.

Garavand et al. [65] proposed innovative composite films based on corn starch/PVA blends (starch:PVA 40:60) and loaded with three different levels of chitosan nanoparticles (CNPs) (1,3 and 5% *w*/*v*). Water vapor transmission test results showed that the WVP values of the starch-PVA nanocomposite films were significantly decreased by adding CNPs into the film matrix. The WVP of neat starch-PVA films was 0.41 g·mm/kPa·h·m^2^ and dropped to 0.28 g·mm/kPa·h·m^2^ due to the addition of CNPs up to 5% *w*/*v*.

Amaregouda et al. [66] conducted surface modifications of CuO nanorods with L-alanine amino acid under microwave radiation to obtain CuO-L-alanine. Different amounts of modified CuO nanorods (2, 4, 6, and 8 wt%) were introduced into the PVA/carboxymethyl cellulose(CMC) blends to prepare PVA/CMC/CuO-L-alanine NC films. When the PVA/CMC matrix was filled with 8% CuO-L-alanine, the tensile strength was increased from 28.58 ± 0.73 to 43.40 ± 0.93 MPa, and the UV shielding ability and the barrier to water vapor were greatly improved. In addition, antioxidant properties and antibacterial properties were obtained.

#### 2.2.2. Chemical Cross-Linking and Polymer Co-Blend Modification

In this method, the cross-linked polymer resulting from the cross-linking of the cross-linking agent with PVA is blended with another polymer, or, the cross-linking agent is cross-linked with both the PVA and the other polymer in the blended polymer.

Chen et al. [67] fabricated a multi-crosslinked biodegradable film containing gelatin (Ge), dialdehyde oligomeric chitosan (DOC), and PVA with excellent mechanical properties. DOC acts as both an antioxidant and antimicrobial component and a cross-linking agent to covalently cross-link the gelatin matrix to form the first gelatin network. PVA cross-linked through its crystalline domains acts as a second network, interpenetrating the first gelatin network. The films with this structure contain multiple cross-links of hydrogen bonds, covalent bonds, and crystalline domains, all of which can act as sacrificial bonds for energy dissipation. As a result, the prepared film has high mechanical strength and toughness. In addition, it exhibits good light transmission, improved moisture resistance, and biodegradability in natural soil with food preservation efficiency comparable to that of commercial PE materials.

Sajjan et al. [68] prepared PVA/Ge/polyethylene glycol-400 (PEG-400) films with biodegradable properties. These films were then cross-linked using formaldehyde to enhance the barrier properties. Fourier transform infrared spectroscopy (FTIR), and wide-angle X-ray diffraction (WAXD) confirmed the cross-linking between the PVA and Ge components, as well as the effect of hydrogen bonding on crystallization. Formaldehyde cross-linking greatly affected the tensile strength and elongation at the break of the films. In addition, it has excellent thermal stability, mechanical strength, low water solubility (10.17–10.89%), low water vapor transmission rate (0.075–0.2272 g/cm^2^ h), and high moisture retention (95.64–96.29%) compared to the uncrosslinked film; it is also biodegradable and can therefore be used as a shopping bag.

Wen et al. [69] successfully prepared a biodegradable food packaging film (PVA/CMCS/CA) with anti-fog and antibacterial properties by blending PVA and carboxymethyl chitosan (CMCS) and adding citric acid (CA) as a cross-linking agent. The hydrogen bonding between PVA and CMCS was confirmed. The tensile strength of the film increased from 21.03 MPa to 29.65 MPa, and the Young’s modulus increased from 3.71 MPa to 10.87 MPa as the CA content reached 5 wt%. CMCS and CA were found to affect the crystallization of the PVA composite film and help promote soil microbial degradation of the film. CA enhanced the cross-linking between PVA and CMCS to form a cross-linked network, which improved the thermal stability of the composite films and reduced their water vapor permeability and swelling.

## 3. Application of PVA Water Resistant Films

There are many applications for PVA-based water-resistant films, mainly in the packaging field and optoelectronic field.

### 3.1. Packaging Field

Packaging is a necessity for commodity circulation, and flexible packaging materials, mainly paper and plastic film, occupy a large proportion in the packaging field [70]. Paper is widely available and has easy processing and high printability; however, the barrier and mechanical properties are poor, and the application has limitations [71,72,73]. PVA water-resistant film has the mechanical properties of general-purpose plastic films, such as PE, PP, PVC, but also has high barrier properties, biodegradable properties, good biocompatibility, and more applicability compared to paper packaging, but is also in line with the “plastic ban” requirements of green development. When used in the packaging field, the components of PVA water-resistant film should meet the basic requirements of packaging materials, and all additives should be non-toxic and non-hazardous [74].

Wang et al. [75] synthesized a bio-nanocomposite using silk protein (SF)/PVA as the matrix, TEMPO-oxidized bacterial cellulose nanofibers (TBC) as the reinforcement, and silver nanoparticles (AgNPs) as the antimicrobial agent. It was found that TBCs and AgNPs were uniformly dispersed in the synthesized nanocomposite, which resulted in excellent mechanical properties of the material. By adjusting the content of TBCs, the mechanical properties, hydrophobicity, and water vapor barrier of the material could be adjusted. In addition, the bio-nanocomposite film had strong UV shielding properties, antibacterial activity, and low silver leakage. These properties make the SF/PVA/AgNPs/TBC bio-nanocomposite promising for active food packaging and seed storage/packaging applications.

Yihun et al. [76] prepared maleated chitin nanofiber (MCNFs)-PVA composite films, and FTIR and XRD analyses showed strong interactions between MCNFs and PVA. The nanocomposite films showed similar transparency levels as the PVA films, indicating that the MCNFs were dispersed in the nanoscale range. Both DSC and TGA analyses showed that the thermal stability of PVA was significantly enhanced by the addition of MCNFs. Tensile test data also showed that the Young’s modulus, tensile strength, elongation at break, and toughness of PVA were increased by 71.87%, 41.47%, 49.10%, and 261.20%, respectively, with the addition of 3 wt% MCNFs to the PVA matrix. For the composite film containing 3 wt% MCNFs, the maximum swelling index of PVA film in deionized water (195%) was reduced to 141%. The strong interfacial interaction between MCNFs and PVA was the decisive factor in improving the performance of PVA. Overall, this study demonstrates a substantial approach to preparing thermomechanically stable, water-insensitive biodegradable nanocomposite films for the manufacture of flexible packaging materials.

Gürler et al. [77] prepared biocomposite films with a combination of starch, CA at different concentrations (3%, 6%, and 9%), and PVA by the flow casting method. The physicochemical properties of the starch-PVA films were improved by adding citric acid to the esterification reaction, and the solubility, swelling capacity, and water vapor permeability decreased with increasing citric acid. The tensile strengths of the starch-PVA-glucose blends containing 3% citric acid (SPG3) and starch-PVA-fructose blends containing 3% citric acid (SFG3) biocomposite films were 10.17and 8.24 MPa, respectively, while the tensile strengths of the starch-PVA-fructose blend without citric acid were 8.16 and 7.94 MPa, respectively. The cell viability values of the biocomposite films of NIH-3T3 and L929 cells were calculated to be 90% compared to the control group, indicating that they were also quite biocompatible. These materials can be used in the field of food packaging.

### 3.2. Optoelectronic Field

PVA has high solubility, low slip, environmental friendliness, transparency, dielectric strength [78,79], and the ability to interact with different fillers through the -OH group of the polymer chain. The properties of PVA make it an important candidate material for optoelectronic devices [80,81,82,83]. Inorganic nanoparticles such as NiO [84], Au, Ag, Ga_2_O_3_ [85], MgO, ZnO [86], and SiO_2_ [87,88,89] have been used as nano-fillers in PVA substrates with unique properties such as high thermal and chemical stability, biocompatibility, non-toxicity, and high surface reactivity [90,91]. It was observed that the thermal and mechanical properties of the resulting PVA composite films were enhanced, and the optical and dielectric properties were improved [92].

El-Zahhar et al. [93] prepared PVA/lead iodide (PVA/PbI_2_) composite films by adding lead iodide to PVA. Tests showed that the crystalline properties increased with increasing PbI_2_ content, and SEM micrographs revealed a uniform particle distribution of PbI_2_ within the polymer. The optical parameters of the PVA/PbI_2_ films were measured by UV-Vis absorption-transmission spectroscopy. The indirect energy gap of PVA is 4.75 eV, while the PVA/PbI_2_ film with 10 wt% PbI_2_ shows a direct jump of 2.47 eV and an indirect jump of 2.15 eV. The resistivity of the PVA film was 1.79 × 109 Ω cm in the dark and decreased to 1.85 × 108 Ω cm at 7000 lux. The film resistivity was found to decrease as the PbI_2_ content ratio increased to 10 wt%. The results also indicate that the photosensitivity gradually increases as the PbI_2_ content ratio in the composite film increases. The optical properties suggest that the composite films (PVA/PbI_2_) can be used for different optoelectronic applications.

Kalyani et al. [94] prepared nonporous PVA/MgO nanocomposite films using solution flow-delay technique by dispersing 5, 10, and 15 wt% of MgO filler in 95, 90, and 85 wt% of PVA matrix, respectively. It was confirmed that the -OH group of PVA interacts with Mg-O through hydrogen bonding, and optical studies showed that the absorbance of the nanocomposite films increased with increasing MgO, while the bandgap energy decreased with increasing MgO loading. EIS studies showed that the ionic conductivity of the composite films increased with the increase of filler content. The permittivity and dielectric constant (ε’) of the PVA@MgO nanocomposite films decreased with increasing frequency and increased with increasing MgO wt% in the PVA matrix. The addition of MgO to PVA to obtain PVA@MgO films is an indispensable candidate for modern industrial/engineering applications. MgO in PVA brings meaningful changes in the optical and dielectric properties of the composite films, which may have applications in semiconductor electronics, power electronics, and optoelectronics.

## 4. Summary and Prospect

Globally, a “plastic ban” is under promotion, and the development of PVA film is bright. In order to expand the scope of application of PVA films, it is necessary to modify them for water resistance. The use of multiple modification methods has become a trend for single nano-fill modification, polymer co-blend modification, and chemical cross-linking modification. Because a variety of modification methods use synergistic modification, to further enhance the performance of the film, it will be enhanced; at the same time, the choice of additives will also become greater, leading to more functional films, and thus substantial increases in applicability. For the selection of additives, to integrate environmental protection, processing costs, and many other factors, for example, in the field of packaging, to consider the compatibility of blended polymers and PVA matrix, toxicity, degradability, preferably starch, gelatin, chitosan, and other biomaterials, for the choice of nanoparticles, nanocellulose, nano-chitosan, and other biomaterials can be preferentially selected.

PVA water-resistant film applications will become more and more extensive, thanks to the development of additives. The future direction of the development of additives should be green, pollution-free, degradable, and low cost. The preparation of safe, biodegradable films requires more research. At the same time, the production process is limited by the inability to efficiently produce large-format films, and the development of new processes for producing large-format films should be the direction of development.

## Figures and Tables

**Figure 1 membranes-12-00347-f001:**
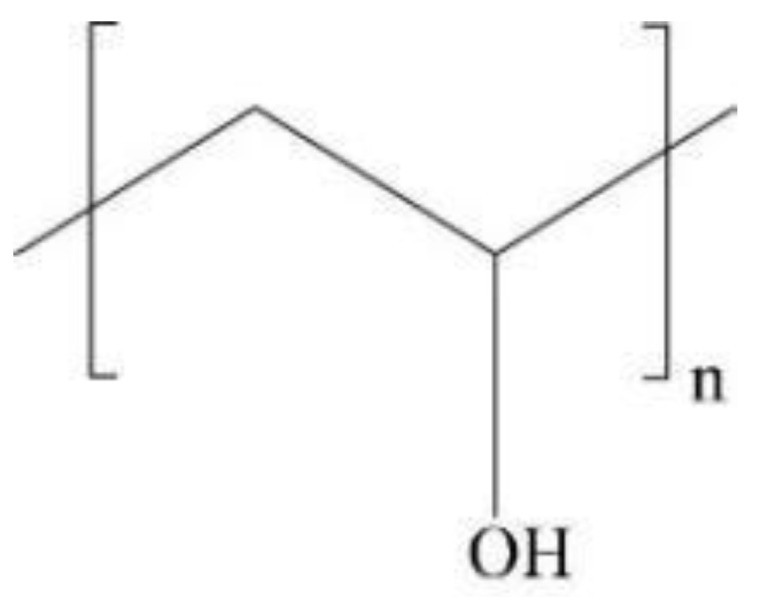
The structural formula of PVA.

**Figure 2 membranes-12-00347-f002:**
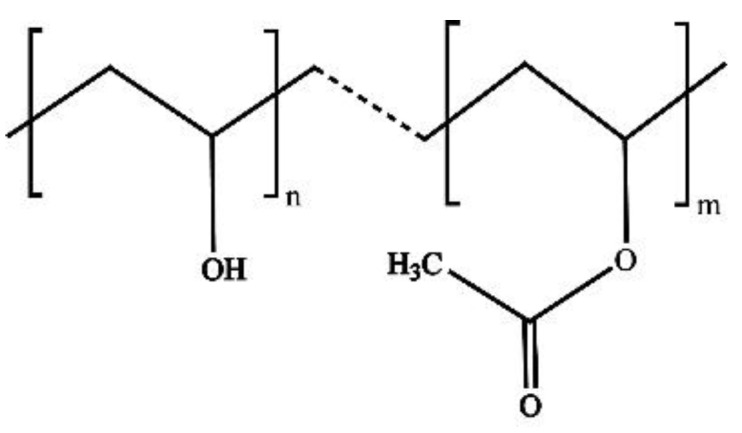
Schematic diagram of PVA and PVAc coexistence.

**Figure 3 membranes-12-00347-f003:**
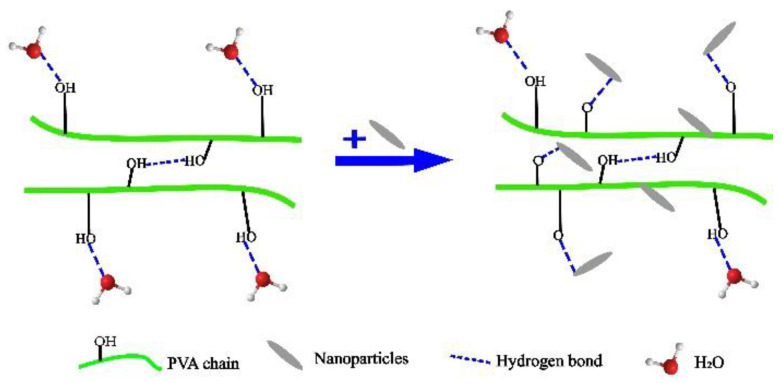
Diagram of nano-filling modification.

**Figure 4 membranes-12-00347-f004:**
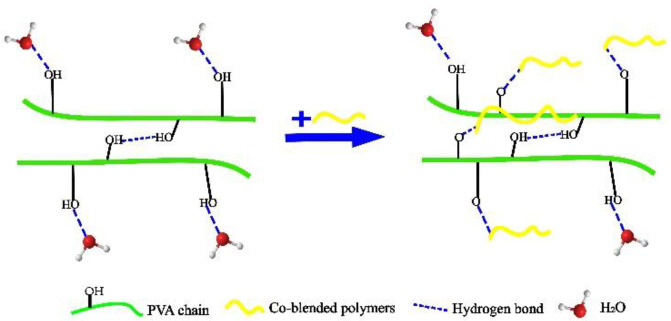
Diagram of polymer co-blend modification.

**Figure 5 membranes-12-00347-f005:**
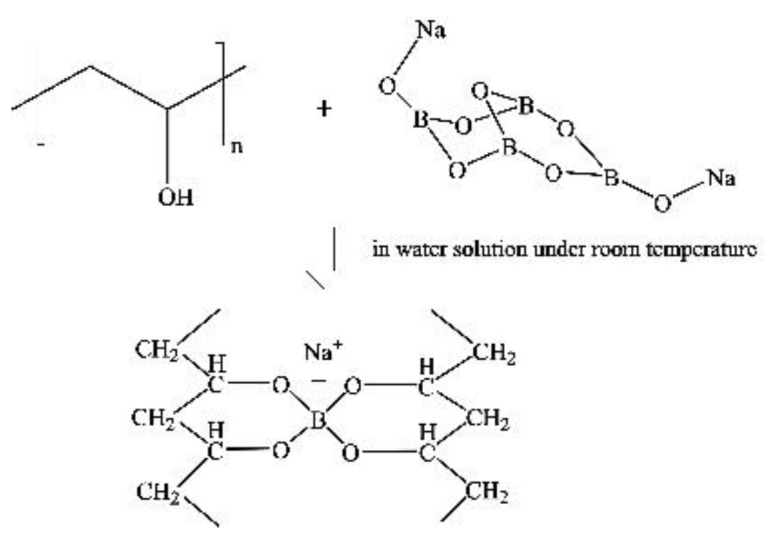
Diagram of the cross-linking reaction between boric acid and PVA.

**Figure 6 membranes-12-00347-f006:**
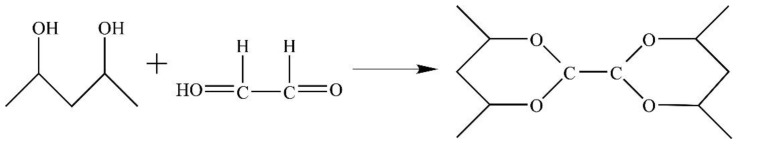
Diagram of the cross-linking reaction between glyoxal and PVA.

**Figure 7 membranes-12-00347-f007:**
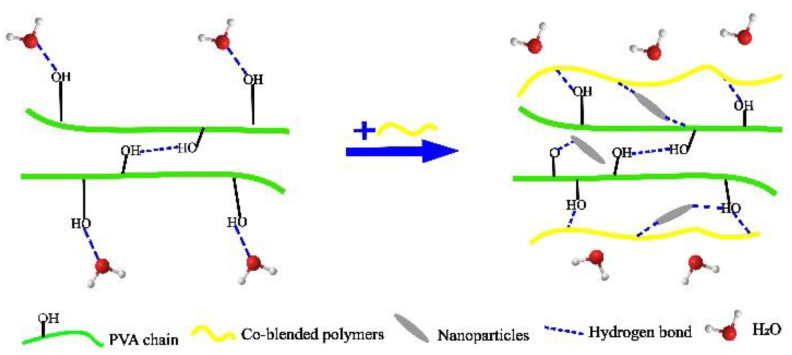
Diagram of the nanofiller and polymer co-blend acting together.

## Data Availability

Not applicable.

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
