# Peer review of "Research Progress of Polyvinyl Alcohol Water-Resistant Film Materials"

_membranes, 2022, doi:10.3390/membranes12030347_

Round 1

Reviewer 1 Report

Accept...

Reviewer 2 Report

As the authors addressed all of the reviewers' recommendations and improved their work, I propose that it be published in Membrane in its current version.

This manuscript is a resubmission of an earlier submission. The following is a list of the peer review reports and author responses from that submission.

Round 1

Reviewer 1 Report

Major comments

(1) The figures amended are not proportionate to the review article. 
(2) The literature survey on this material is plenty and not covered in this review article.
(3) It would be meaningful if similarities and differences among the cited reports are emphasized.

Reviewer 2 Report

The paper is interesting. However, there is a lack of consistency in writing. Followings are suggested to improve the consistency of the manuscript. The composition of the modified PVAs (composites) along with the quantified enhancement of water resistance as compared to only PVA shall enhance the quality of the manuscript.  Some elaboration is made as follows:

Title: Seems not OK but the text is also discussing the other parameters like toughness and strength of the composite materials apart from only  water barrier characteristics.

Abstract: Brief and comprehensive.

Keywords: OK

Introduction:

Line 34 needs a reference.

Line 73-78 strongly need a reference.

Line 80-83: Correct the grammar of the sentence.

Line 94-98: What was the composition of the composite material and how much water resistance was enhanced with the modificatio?

Line 99-105: What was the composition of the composite material? Also, describe as what was the mode of combination? Was it some reinforcement? The mode of combination and the materials compositions are clearly described in section 2.1.2. But in 2.1.1, these are missing at large in the 2nd paragraph.

It would be better if you also indicate the polymer (I think it is chitosan), which has been blended.

Line 128-136: The main emphasis of the paper is the water-resistance with PVA. However, the paragraph presents this particular parameter with a very low weightage. Kindly highlight the focused parameter i.e. water resistance more prominently.

Section 2.1.3: There is no need to repeat “The side chains of PVA have a large number of hydrophilic groups hydroxyl groups, which exhibit a strong affinity for water”. This has already been described.

Section 2.2: Multiple synergistic: I personally think both have the same meaning. The start of section 2.2.1 is somewhat awkward, starting directly from a picture.

Line 191: Kindly confirm (W/v) whether it is the same i.e. both weight and volume.

Line 189-199: Kindly confirm they also enhanced the water barrier properties as this is the main focus of your paper.

Line 336: Kindly avoid second-person pronouns like “you”.

Synthesis is common in sections 2 and 3. Whereas, the flow of the text suggests that section 2 is dedicated to the synthesis and section 2 is dedicated to only applications.

Reviewer 3 Report

  • Because it is a review paper, the number of references should be more. There is a lot of relevant work that should be explored, and here are some suggested sources: Materials Chemistry and Physics(2022): 125731; Journal of Inorganic and Organometallic Polymers and Materials10 (2021): 4141-4149; Current Nanoscience 16.6 (2020): 994-1001; Chemical Engineering Research and Design 171 (2021): 268-276; Advanced Energy Materials 10.18 (2020): 1903951.
  • Furthermore, there are other sources that are not acknowledged in the introduction.
  • There are several blank areas throughout the document.
  • Tables comparing other published related studies should be included in the article.
  • The review overlooks the morphology, structural, physical, and chemical aspects of PVA, as well as its impact on water treatment.